# Effects of Liposome and Cardiolipin on Folding and Function of Mitochondrial Erv1

**DOI:** 10.3390/ijms21249402

**Published:** 2020-12-10

**Authors:** Xiaofan Tang, Lynda K Harris, Hui Lu

**Affiliations:** 1School of Biological Sciences, Faculty of Biology, Medicine and Health, The University of Manchester, Manchester M13 9PT, UK; xf.tang@siat.ac.cn; 2Department of Materials, Faculty of Science and Engineering, The University of Manchester, Manchester M13 9PT, UK; 3Shenzhen Engineering Laboratory of Nanomedicine and Nanoformulations, Shenzhen Institutes of Advanced Technology (SIAT), Chinese Academy of Sciences, Shenzhen 518055, China; 4Division of Pharmacy and Optometry, School of Health Sciences, Faculty of Biology, Medicine and Health, The University of Manchester, Manchester M13 9PL, UK; Lynda.K.Harris@manchester.ac.uk; 5Maternal and Fetal Health Research Centre, Faculty of Biology, Medicine and Health, The University of Manchester, Manchester M13 9WL, UK; 6Maternal and Fetal Health Research Centre, Manchester University NHS Foundation Trust, Manchester Academic Health Sciences Centre, St Mary’s Hospital, Manchester M13 9WL, UK

**Keywords:** sulfhydryl oxidase, MIA pathway, mitochondrial membrane, enzyme kinetics

## Abstract

Erv1 (EC number 1.8.3.2) is an essential mitochondrial enzyme catalyzing protein import and oxidative folding in the mitochondrial intermembrane space. Erv1 has both oxidase and cytochrome *c* reductase activities. While both Erv1 and cytochrome *c* were reported to be membrane associated in mitochondria, it is unknown how the mitochondrial membrane environment may affect the function of Erv1. Here, in this study, we used liposomes to mimic the mitochondrial membrane and investigated the effect of liposomes and cardiolipin on the folding and function of yeast Erv1. Enzyme kinetics of both the oxidase and cytochrome *c* reductase activity of Erv1 were studied using oxygen consumption analysis and spectroscopic methods. Our results showed that the presence of liposomes has mild impacts on Erv1 oxidase activity, but significantly inhibited the catalytic efficiency of Erv1 cytochrome *c* reductase activity in a cardiolipin-dependent manner. Taken together, the results of this study provide important insights into the function of Erv1 in the mitochondria, suggesting that molecular oxygen is a better substrate than cytochrome *c* for Erv1 in the yeast mitochondria.

## 1. Introduction

Mitochondria are important organelles in eukaryotic cells, which produce most of the energy for biological processes. Since about 99% of proteins in mitochondria are imported from the cytosol, mitochondria have dedicated protein import systems for mitochondrial protein biogenesis [1]. Protein import and oxidative folding, conjugated with disulphide bond formation, is a unique protein import mechanism in the mitochondria intermembrane space (IMS), which is executed by the mitochondrial import and assembly (MIA) pathway [2,3,4,5]. Erv1 (essential for respiration and viability) [1] in yeast, or called ALR (augmenter of liver regeneration) in mammals, is an essential component of the MIA pathway, which works together with Mia40 to catalyze the oxidative folding of the newly imported substrate proteins in the IMS [6,7,8]. Mia40 acts as an oxidoreductase, interacting with the substrate proteins and transferring disulphide bonds to the substrates. Erv1, a FAD-binding protein, serves as a disulphide bond generator to reoxidize the reduced Mia40. Then, the reduced Erv1 is regenerated by passing electrons to molecular oxygen (acting as an oxidase) or cytochrome *c* (as cytochrome *c* reductase) via the flavin cofactor (Figure 1A) [9,10,11].

Erv1 belongs to the Erv/ALR sulfhydryl oxidase family; the homologue proteins have been identified in all mitochondria-containing eukaryotes [12,13,14]. They all have a highly conserved catalytic domain of ~100 amino acids folded in a helix bundle and form stable dimers by noncovalently binding to flavin adenine dinucleotide (FAD) (Figure 1B) [15,16,17]. The catalytic (or FAD-binding) domain of yeast (Saccharomyces cerevisiae) Erv1 is at the C terminus, containing a redox active site CXXC disulphide bond (Cys130-Cys133) and a CX16C structural disulphide (Cys159-Cys176) [18,19,20]. Yeast Erv1 also has a functionally important shuttle disulphide bond (Cys30-Cys33) in the non-conserved and unfolded N-terminal domain, which is functionally important for interaction with Mia40 [18,20].

Erv/ALR enzymes can act as an oxidase by passing electrons to molecular oxygen and reducing oxygen to hydrogen peroxide, or as a cytochrome *c* reductase by passing electrons to oxidized cytochrome *c* (Figure 1A). Enzyme kinetic studies revealed that different Erv1/ALR enzymes have different oxidase and cytochrome *c* reductase activities [17,21,22,23]. While both yeast Erv1 [23] and human ALR [21] seem to prefer cytochrome *c* more than molecular oxygen as an electron acceptor, the ratio of relative cytochrome *c* reductase to oxidase activity is quite different for these enzymes, with 15 for yeast Erv1 [23] and 107 for human ALR [21], as ALR uses oxygen poorly. On the other hand, yeast Erv2 [17] and avian QSOX [22] are good oxidases and readily reduce oxygen to hydrogen peroxide.

Mitochondria contain two membranes, the outer membrane (OM) and the inner membrane (IM), which are composed of phospholipid bilayers and proteins. The mitochondrial IMS is narrow and is almost filled with the cristae of the mitochondrial inner membrane, due to its large surface area. Biological reactions that take place in the IMS are surrounded by the mitochondrial IM and inevitably would be influenced by the membrane environment. Phosphatidylcholine (PC) and phosphatidylethanolamine (PE) are the major mitochondrial phospholipids in all cell types, which account for about 80% of total phospholipids. Cardiolipin is enriched in the mitochondrial IM of all organisms and may be confined to the IM, as has been argued for plants [24]. The majority of cardiolipin, almost 80%, is located in the mitochondrial IM in yeast cells and constitutes about 20% of the membrane lipids found there [25]. Cardiolipin possesses a unique structure containing four (instead of two) fatty acid tails and two negatively charged phosphate groups, as shown in Figure 2A.

Although cytochrome *c* is a water-soluble protein, it is membrane-associated. Studies have shown that about 85% of cytochrome *c* is stored in the inner membrane cristae, and only 15–20% of total cytochrome *c* is available in the IMS [25,26]. Cardiolipin is the preferential and major binding partner for cytochrome *c* [27], and increasing cardiolipin in liposomes caused increased cytochrome *c* binding [28]. Binding of cytochrome *c* to membranes also led to an increase in peroxidase activity but was without effect on protein conformation or membrane penetration [29,30]. These findings suggest that the presence of membranes may play an important role in the folding and activity of these proteins.

Moreover, mitochondrial subfractionation analysis showed that about 50% of Erv1 [31] and ALR [32] are peripherally associated with the mitochondrial IM. However, whether this membrane association affects the folding, and, hence, function of these proteins, is unknown. Here, in this study, we investigated whether membrane association and the presence of cardiolipin play a role in mediating the oxidase and cytochrome *c* reductase activities of Erv1, using liposomes as model bio-membranes to mimic the mitochondrial membrane. Our results showed that the presence of liposomes inhibits the cytochrome *c* reductase activity of Erv1 strongly and in a cardiolipin concentration-dependent manner. The results of this study suggest that molecular oxygen is a better substrate than cytochrome *c* for yeast Erv1 in the mitochondria.

## 2. Results

### 2.1. Preparation and Characterization of Liposomes

The major components of the yeast mitochondrial OM are phosphatidylcholine (~45%), phosphatidylethanolamine (~23%), cardiolipin (~6%), and other lipids (~26%). For the IM, they are phosphatidylcholine (38%), phosphatidylethanolamine 30%), cardiolipin (20%), and other lipids (12%) [33]. The acyl chains of phospholipids of the yeast *S. cerevisiae* are mainly C16:1 and C18:1 with minor amounts of C14:0, C16:0, and C18:0 [34]. Since C16:1 lipids are not commercially available, liposomes with a lipid composition that resembles the mitochondrial membranes were prepared using C18:1 lipids, 1,2-dioleoyl-sn-glycero-3-phosphocholine (DOPC), and 1,2-dioleoyl-sn-glycero-3-phosphatidylethanolamine (DOPE) at a fixed ratio of 5:3 and in the presence of 0.5% or 15% of cardiolipin (Figure 2). We named these liposome formulations CL0, CL5, and CL15, accordingly, in this report. The average diameter of the liposomes in each formulation was measured using dynamic light scattering (Figure 2B) and was found to be the same, with or without the presence of cardiolipin. The mean diameter was 160 ± 1 nm, which was as expected, based on previous results [35].

Lipid packing is a crucial feature for liposomes, which can be analyzed using polarity-sensitive probes whose emission spectra depends on lipid packing. Laurdan, a microenvironment-sensitive fluorescent probe, was used to characterize the liposomes and assess whether the presence of cardiolipin affected the liposome packing, as described previously [27]. The fluorescence intensity of Laurdan at 440 and 490 nm (with excitation at 340 nm) was sensitive to the ordered and disordered lipid phases, respectively [27]. As shown in Figure 2C, the fluorescence spectra of Laurdan in all three liposomes showed a dominant peak at 440 nm. The overall spectra of Laurdan in CL0 and CL5 liposomes were very similar, suggesting that the lipids in both formulations were in a homogeneously ordered phase. However, a broad peak at around 490 nm appeared in the CL15 liposome spectrum, suggesting the presence of a disordered lipid phase. Thus, the general polarity (GP) of the liposomes was calculated based on: GP = (I440 − I490)/(I440 + I490). The calculated GPs varied from 0.36 for CL0, 0.33 for CL5, and to 0.13 for CL15, indicating that the structure of the liposomes changed from an ordered phase toward a disordered phase. The polarity decreased by about 60% with the presence of 15% cardiolipin in CL15 compared with CL0, suggesting that cardiolipin significantly changes the structure of liposomes, possibly because cardiolipin is large in size, with four alkyl tails that disrupt the lipid organization.

### 2.2. Effects of Liposomes on the Oxidase Activity of Erv1

It was reported that approximately 50% Erv1 is peripherally associated with the mitochondrial IM [31]. However, whether the membrane association affects Erv1 folding and/or activity is unknown and unreported. Liposomes are a good model in which to study kinetic properties of enzymes in a membrane environment. Here, we investigated how the presence of the CL0, CL5, and CL15 liposome formulations affects the folding and oxidase activity of Erv1.

Erv1 is a FAD-binding protein with a signature UV-vis absorption spectrum including a peak maximum at 460 nm. The absorption spectra of Erv1 in the absence and presence of the various liposome formulations were almost identical (Figure 3), suggesting the presence of these liposomes did not affect Erv1 FAD binding.

Next, the effects of the liposomes on the Erv1 oxidase activity were investigated using oxygen consumption analysis (Figure 4A), as described previously [23,36]. As shown at the beginning of Figure 4A, there was no oxygen consumption by DTT in the absence of Erv1. The oxygen consumption curves were differentiated, the oxygen consumption rate of oxygen against oxygen concentration was plotted, and the enzyme kinetic parameters K_m_ and k_cat_ were estimated (Figure 4B, Table 1). The data showed that addition of liposomes decreased both K_m_ and k_cat_ to a similar level (Figure 4C). Overall, the relative oxidase catalytic activity of Erv1 was reduced to about 70% of the control (in the absence of liposomes) in all three cases (Figure 4D). Thus, the presence of liposomes suppressed the oxidase activity of Erv1 partially, regardless of cardiolipin content, suggesting that cardiolipin was not a specific binding partner for Erv1.

### 2.3. Effects of Liposomes on Cytochrome c Reductase Activity of Erv1

Since cytochrome *c* is closely associated with the mitochondrial IM, whether heme binding to cytochrome *c* was affected by liposomes was assessed first. For this, UV-vis absorption spectra were recorded in the presence and absence of the liposomes. As shown in Figure 5A, the spectra were almost the same, suggesting cofactor binding to cytochrome *c* was unaffected and no conformational changes were caused by the liposomes.

Next, we investigated whether the presence of liposomes affects the cytochrome *c* reductase activity of Erv1, using DTT as an electron donor and cytochrome *c* as an acceptor, as established previously for Mia40 [18,23]. Time courses of cytochrome *c* absorbance change at 550 nm were measured in the absence and presence of various liposome formulations, which showed that the absorbance change was decreased by the presence of liposomes and was dependent on cardiolipin concentration (Figure 5B). In all cases, the rate of non-enzyme catalyzed cytochrome *c* reduction by DTT in the absence of Erv1 was negligible. The absorbance changes were analyzed by differentiation, and the enzyme parameters were calculated (Figure 5C, Table 1). While the initial rate of cytochrome *c* reduction was decreased to about 90% in the presence of the CL0 formulation, it was decreased further to about 74% and 58% of that of the control (without liposomes) in the presence of CL5 or CL15 liposomes (Figure 5C). The overall catalytic efficiency (k_cat_/K_m_) of cytochrome *c* was reduced to 40–10% in the presence of CL0–CL15 (Figure 5D, Table 1).

Taken together, our results showed that the liposomes have a strong inhibitory effect on the cytochrome *c* reductase activity of Erv1 and that this inhibition increased with the increasing cardiolipin content. With the presence of 15% cardiolipin in the liposome, a concentration similar to that observed in the mitochondrial IM, the cytochrome *c* reductase activity of Erv1 was reduced to about 10% that of the control (in the absence of liposomes). This was largely due to the effect of CL15 on K_m_, the binding of cytochrome *c* to Erv1.

## 3. Discussion

In this study, we investigated how liposomes and cardiolipin affect the oxidase and cytochrome *c* reductase activities of mitochondrial Erv1. It was reported that about 50% Erv1 was peripherally associated with the mitochondrial IM [31]. The overlapping spectra and slightly decreased oxidase activity of Erv1 at the molar ratio of lipids:Erv1 between 60:1 to 100:1 suggested that Erv1 was probably not bound to liposomes. Considering the high abundance of proteins in mitochondrial IMS and the localization of Erv1′s partner protein Mia40, the membrane association of Erv1 in mitochondria is most likely due to its interaction with Mia40, rather than lipids.

Unlike the oxidase activity, the liposomes had a much clearer impact on the cytochrome *c* catalytic efficiency of Erv1, and the inhibitory effect increased with the increased cardiolipin content of the liposomes (Figure 5). Our results suggested that cardiolipin, PC, and PE inhibit the cytochrome *c* reductase activity of Erv1 by affecting the interactions between Erv1 and cytochrome *c* since the K_m_ increased with the presence of the liposomes and in a cardiolipin concentration-dependent manner. Cardiolipin was predominantly responsible for the inhibitory effect. Cardiolipin has four acyl chains, equivalent to two PC or PE molecules (see Figure 1A). Furthermore, there are two negative charges on the head group of cardiolipin, which favors electrostatic attractions with the highly positive-charged cytochrome *c*, which has a pI of about 9. Consequently, the interactions between Erv1 and cytochrome *c* weakened, as evidenced by the 4.6-fold increase in K_m_, compared with the K_m_ of the control and CL15 (Table 1). Moreover, the addition of cardiolipin changed the liposome structure from an ordered to a more disordered phase in CL15. Thus, we would predict that CL15 liposomes would have a less rigid structure than orderly packed liposomes, which may enhance cristae formation and favor the interaction of cytochrome *c* with the liposomes or mitochondria IM. Hence, the interactions or communications between cytochrome *c* and Erv1 were weakened or partially reduced.

The presence of liposomes, such as CL15, decreased the oxidase activity of Erv1 by about 30%, to about 70% of the catalytic efficiency of Erv1 in the absence of liposomes. However, the CL15 formulation inhibited about 90% of the cytochrome *c* reductase activity of Erv1 (Figure 4 and Figure 5, Table 1). We showed that, in the absence of liposomes, the cytochrome *c* reductase activity of Erv1 was slightly higher than that of the oxidase activity, with a relative substrate specificity (RSS), defined as the ratio of (k_cat_/K_m_)^Cyt c^ to (k_cat_/K_m_)^Oxygen^ between 2–15, depending on the experimental conditions [23] (Table 1). In this study, our results showed that in the presence of CL15 liposomes, the RSS ratio is about 0.33 (Table 1). Thus, molecular oxygen is a better substrate than cytochrome *c* in the presence of CL15 liposomes. Since CL15 liposomes mimic the mitochondria IM, this result indicates that molecular oxygen may be also a better substrate than cytochrome *c* in the mitochondrial IMS in yeast. Consistent with this, a previous study showed that the apparent K_m_ of the GSH-Mia40-Erv1 system for oxygen is very low (~3 μM) [23]. About 15% of mitochondrial cytochrome *c* is tightly bound to the cristae of the mitochondrial IM and about 85% of cytochrome *c* is accessible in the bulk solution [37]. However, the concentration of the oxidized cytochrome *c* in the mitochondrial IMS that is available for Erv1 is unknown and it may vary widely depending on cell growth conditions. The results of this study support that molecular oxygen is a better substrate than cytochrome *c* for yeast Erv1 in the mitochondria.

The oxidase activity of Erv1 produces two molecules of hydrogen peroxide (H_2_O_2_) for every one molecule of O_2_ consumed. Hydrogen peroxide is considered a reactive oxygen species (ROS), which, at high levels, can damage all biomolecules and cause oxidative stress. Oxidative stress has been linked to many diseases and to the process of aging [38]. On the other hand, ROS produced under normal physiological conditions play an important regulatory role in cellular metabolic processes, and mitochondria are the main endogenous source of cellular ROS, as a result of incomplete electron transfer along the electron transport chain (ETC) [39]. To avoid H_2_O_2_ accumulation and potential oxidative stress caused by Erv1, yeast has an enzyme, cytochrome *c* peroxidase (Ccp1), in the mitochondrial IMS, which can effectively remove the H_2_O_2_ produced by Erv1 [11]. Under oxidative stress conditions, H_2_O_2_ can also be removed by the IMS-localized glutathione peroxidase Gpx3 [40]. The importance of the Erv1/ALR enzymes was also demonstrated by the identification of a disease directly related to a single mutation in ALR, the human homologue of Erv1 [41]. A single arginine-to-histidine substitution (R194H) in the C-terminus of ALR was identified in children with autosomal recessive myopathy; the effects included respiratory chain deficiency, abnormal mitochondrial morphology, and increased accumulation of mitochondrial DNA (mtDNA) deletions, a temperature-sensitive growth phenotype and reduced cytochrome *c* oxidase (complex IV) activity [41]. Protein characterization studies showed that the mutation results in decreased FAD-binding and thermal stability of the protein, and a mild effect on its function in vitro in the absence of liposomes [20,42]. A study showed that the human short form of ALR exhibited a higher cytochrome *c* reductase activity than its oxidase activity in the absence of liposomes, compared with yeast Erv1 [21]. Moreover, no yeast homologue of Ccp1 has been identified in human. It will, therefore, be interesting to interrogate how liposomes and cardiolipin affect the function of human ALR and, more importantly, the physiological effects. The results of this study showed that molecular oxygen is a suitable substrate for the MIA pathway in yeast. Thus, it will be interesting to understand whether the resulting hydrogen peroxide plays a role in mitochondria–nuclear communication linking mitochondrial oxidative stress status and expression of nuclear DNA-encoded mitochondrial proteins together.

## 4. Materials and Methods

All chemicals used in this study were analytical grade and were from Sigma or Fisher, unless specified. All solutions were prepared using MilliQ water. All experiments were carried out in a Tris buffer (50 mM Tris-HCl, 150 mM NaCl, 1 mM EDTA, pH 7.4) unless specified.

### 4.1. Protein Expression and Purification

The ERV1 gene was cloned using a pET-24a(+) plasmid and expressed in Escherichia coli Rosetta-gamiTM2 cells (Novagen). Cells were grown in Luria broth medium (Formedium Ltd.) containing 50 μg/mL of kanamycin (Fisher Scientific) at 37 °C, until the OD600 reached 0.6. Then, proteins were induced by addition of 0.5 mM isopropyl β-d-1 thiogalactopyranoside (IPTG) at 16 °C and were incubated overnight (16 h). Cells were collected by centrifugation (4 °C, 5000× *g* rpm, 20 min).

Collected cells were resuspended in binding buffer (20 mM imidazol, 50 mM Tris-HCl, 150 mM NaCl, pH 7.4) to ~0.2 mg/mL with 100 μM FAD and two EDTA-free protease inhibitor cocktail tablets (Roche). The cell membranes were ruptured by sonication (10/50 s ON/OFF, 30% amplitude, 50 min) and the cell debris and soluble proteins were separated by centrifugation (4 °C, 17,000 rpm, 60 min). The supernatant containing the majority of the expressed proteins was carefully collected and filtered with a 0.45-μm filter. Then, the supernatant was purified following a standard protocol for His-tagged protein purification. (1) Firstly, the supernatant was applied to a column packed with a 2-mL volume of Ni^2+^ charged Hi Ni-NTA (Ni^2+^-nitrilotriacetate) His-Bind beads (Novagen). (2) Following binding of the target proteins, the beads were washed with 10 mL binding buffer. (3) The protein-bound beads were then washed with 20 mL of washing buffer (40 mM imidazol, 50 mM Tris-HCl, 150 mM NaCl, pH 7.4). (4) The target proteins were eluted by washing with 6 mL elution buffer (500 mM imidazol, 50 mM Tris-HCl, 150 mM NaCl, pH 7.4) and stored in an excess of FAD at −80 °C for preservation.

Cytochrome *c* (cat.9007-43-6) was purchased from Sigma and its redox state was checked, based on absorption at 550 nm; more than 95% of the proteins were in their oxidized form and used as purchased.

### 4.2. Liposome Preparation

Liposomes were prepared by dissolving 1,2-dioleoyl-sn-glycero-3-phosphocholine (DOPC; 500 μmol/L), 1,2-dioleoyl-sn-glycero-3-phosphocholine (DOPE; 300 μmol/L), and varied concentration of cardiolipin (0, 50, 150 μmol/L) in chloroform to mimic the molecular ratio of the mitochondria IM (PC 50%: PE 30%: CL 15%). Solvent was removed by rotary evaporation to produce a thin lipid film, which was dried in a vacuum oven overnight. The lipid mixtures were rehydrated with Tris buffer (50 mM Tris-HCl, 150 mM NaCl, 1 mM EDTA, pH 7.4) to yield a lipid concentration of 1 mM. The resulting suspension was extruded 11 times using a 1-mL Mini-Extruder (Avanti Polar Lipids) through a 0.1-μm, 19-mm polycarbonate membrane, surrounded by two 10-mm filter supports, in order to produce a unilamellar liposome formulation. The suspension was stored at 4 °C until use. The size (hydrodynamic diameter) distributions were measured by dynamic light scattering (DLS).

### 4.3. UV-Visible Spectroscopy

Absorption spectra were recorded from 250 to 700 nm, at 1-nm intervals, in a 1-cm path length quartz cuvette using a Cary 300 Bio UV-Visible spectrophotometer (Varian). An extinction coefficient of 12.3 mM^−1^ cm^−1^ at 460 nm was used to calculate concentration of Erv1, and ε 450 nm of 11.3 mM^−1^ cm^−1^ was used for free FAD. An extinction coefficient ε 550 nm of 29.5 mM^−1^ cm^−1^ for oxidized cytochrome *c*, and 8.4 mM^−1^ cm^−1^ for reduced cytochrome *c* were used.

### 4.4. Oxygen Consumption Assays

Erv1 enzymatic activity toward oxygen was measured using a Clark-type oxygen electrode (Hansatech Instruments) in 0.5 mL of reaction volume at 25 °C in Tris buffer (50 mM Tris-HCl, 150 mM NaCl, 1 mM EDTA, pH 7.4), as previously described [21]. When DTT was used as an electron donor, bovine erythrocyte superoxide dismutase 1 (SOD) (Sigma) was added at 10 units/mL to exclude the potential interference of superoxide ions. For Erv1 oxidase kinetic parameter determination, 5 mM DTT was used, so that the DTT concentration was more than 10-fold in excess of the *O*_2_ concentration (~250 µM). The initial slope of the oxygen consumption curve was used to represent the oxygen consumption rate and was calculated by data differentiation using OriginPro software. At least three independent experimental repeats were performed for each experiment. Data were shown as mean ± SEM (*n* = 3), and differences between groups were statistically analyzed using a Kruskal–Wallis (one-way ANOVA) test.

### 4.5. Cytochrome c Reduction Assay

The cytochrome *c* reduction assay was performed using a Cary 300 Bio UV-Visible spectrophotometer (Varian). Reduction rate was measured via the absorbance change of cytochrome *c* at 550 nm. An extinction coefficient ε 550 nm of 29.5 mM^−1^ cm^−1^ for oxidized cytochrome *c* and 8.4 mM^−1^ cm^−1^ for reduced cytochrome *c* were used. The reduction rate of cytochrome *c* was calculated using a ∆ε550 of 21.1 mM^−1^ cm^−1^, the extinction coefficient difference between reduced and oxidized cytochrome *c*. The rate of non-enzyme catalyzed cytochrome *c* reduction by DTT was measured in the absence of Erv1, which was used to calculate the background rate and was subtracted from the rate of reduction, in the presence of Erv1. The data were analyzed using OriginPro software. At least three experimental repeats were performed for all experiments. Data were shown as mean ± SEM (*n* = 3), and differences between groups were statistically analyzed using a Kruskal–Wallis (one-way ANOVA) test.

## Figures and Tables

**Figure 1 ijms-21-09402-f001:**
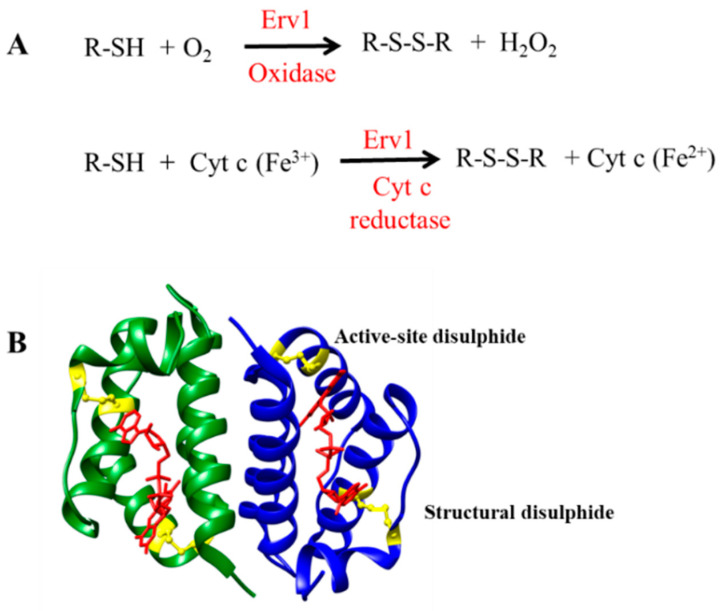
Function and structure of Erv1. (**A**) The reactions catalyzed by Erv1 as an oxidase and a cytochrome *c* reductase, respectively. (**B**) X-ray structure of the C-terminal domain of Erv1. The active site disulphide and the structural disulphide are shown in yellow and FAD is shown in red.

**Figure 2 ijms-21-09402-f002:**
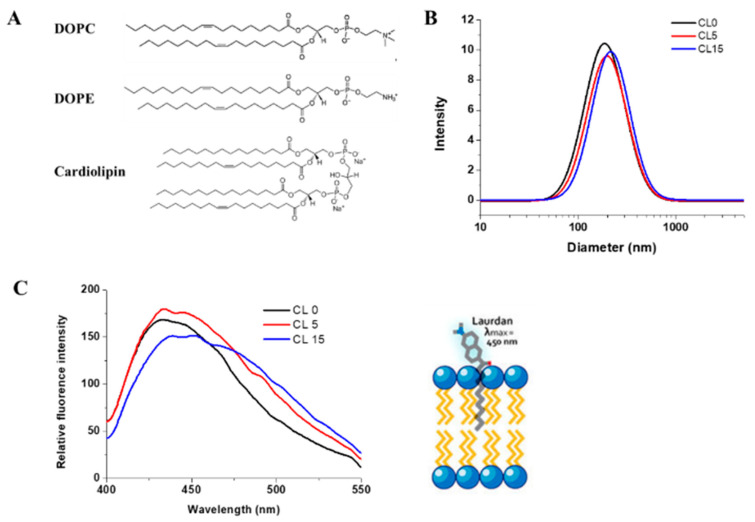
Characterization of liposomes. (**A**) Chemical structures of the lipids used for preparing liposomes. (**B**) Dynamic light scattering spectra of liposomes in the presence of 0%, 5%, and 15% cardiolipin, respectively. (**C**) Fluorescence spectra of Laurdan in each formulation and a schematic model representing the insertion of Laurdan in liposomes. Experiments were performed with 100 µM lipid and 1 µM Laurdan. Lines represent liposome formulations CL0 (black), CL5 (red), CL15 (blue) respectively.

**Figure 3 ijms-21-09402-f003:**
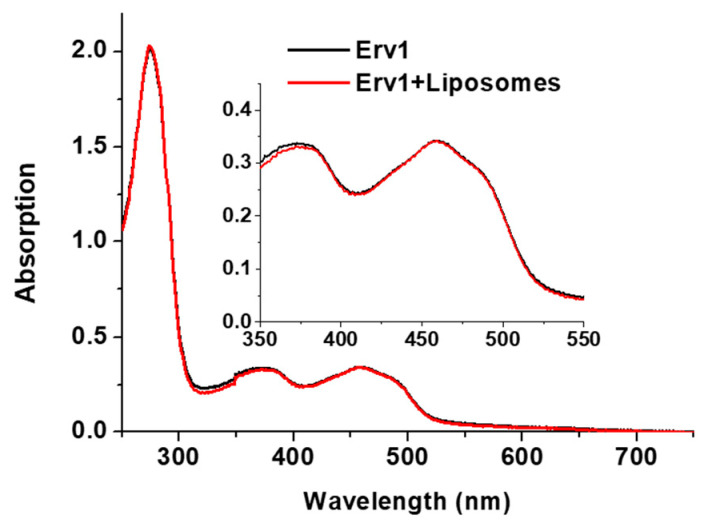
UV-vis absorption spectra of Erv1 in the absence (black) and presence (red) of 0.5 mM CL15 liposomes. The same overlapping spectra were obtained for Erv1 in the presence of CL0 or CL5 liposomes (data not shown). The molar ratio of lipids:Erv1 was 60:1.

**Figure 4 ijms-21-09402-f004:**
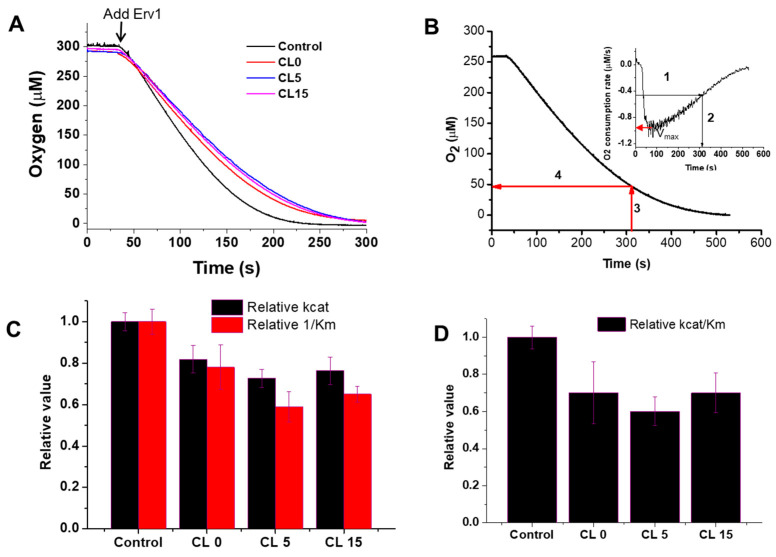
Effects of liposomes on the oxidase activity of Erv1. (**A**) Time course of oxygen consumption by 5 mM DTT catalyzed by 5 µM Erv1, in the absence (Control) and presence of 0.5 mM liposomes CL0, CL5, and CL15, respectively. The molar ratio of lipids:Erv1 was 100:1. (**B**) An example of how the oxygen consumption curve and the first derivative plot (insert) were used to identify the time point at which 50% of the maximal velocity (V_max_) was reached; this was then used for K_m_ estimation. (**C**) The relative oxidase k_cat_ (black) and 1/K_m_ (red) of Erv1. (**D**) Relative oxidase catalytic efficiency of Erv1. Control: without liposomes. CL0, CL5, and CL15: 0.5 mM liposomes. The error bars represent the standard error of the mean (SEM), *n* = 3; differences between each result and the corresponding controls were statistically analyzed using Kruskal–Wallis (one-way ANOVA) test, all *p* < 0.05.

**Figure 5 ijms-21-09402-f005:**
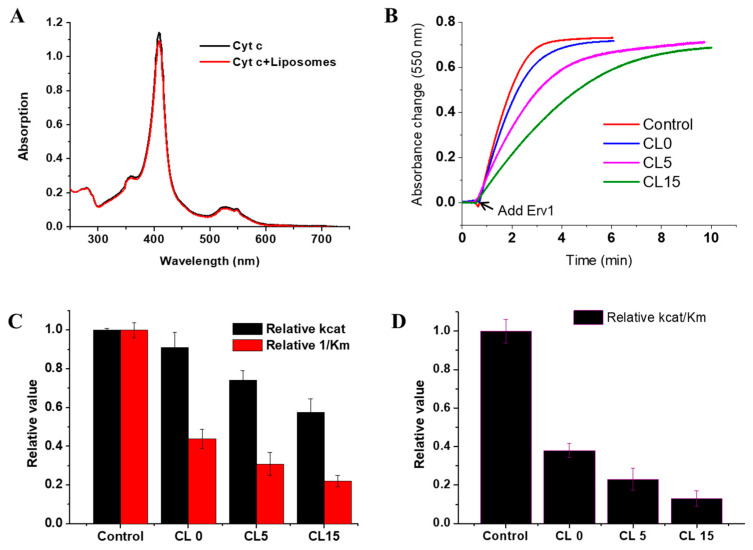
Effects of liposomes on the cytochrome *c* reductase activity of Erv1. (**A**) Absorption spectra of cytochrome *c* in the absence and presence of CL15 liposomes. The same spectra were obtained with CL0 and CL5 liposomes (data not shown) (**B**) Time course of cytochrome *c* reduction, followed by absorbance at 550 nm. (**C**) Relative enzyme kinetic parameters of cytochrome *c* reductase activity, k_cat_ (black) and relative 1/K_m_ (red). (**D**) The relative cytochrome *c* reductase catalytic efficiency of Erv1. Control: the reaction without liposomes. DTT (1 mM) and liposomes (0.5 mM) were used in all cases. The error bars represent the SEM, *n* = 3; differences between each result and the corresponding controls were statistically analyzed using Kruskal–Wallis (one-way ANOVA) test, all *p* < 0.001.

**Table 1 ijms-21-09402-t001:** Summary of the kinetic parameters determined in various conditions, at 25 °C, pH 7.4.

Substrate	Liposomes	k_cat_ (s^−1^)	K_m_ (µM)	k_cat_/K_m_ (M^−1^ s^−1^)
Oxygen	Control (No liposomes)	1.0 ± 0.1	50 ± 10	2.0 × 10^4^ (100%)
	CL0	0.82 ± 0.02	60.0 ± 6	1.4 × 10^4^ (70%)
	CL5	0.74 ± 0.01	58.8 ± 6	1.2 × 10^4^ (60%)
	CL15	0.76 ± 0.02	53.2 ± 5	1.4 × 10^4^ (70%)
Cytochrome *c*	Control (No liposomes)	0.82 ± 0.07	23 ± 4	3.6 × 10^4^ (100%)
	CL0	0.75 ± 0.03	52 ± 4	1.4 × 10^4^ (39%)
	CL5	0.61 ± 0.04	74 ± 3	8.2 × 10^3^ (23%)
	CL15	0.48 ± 0.02	105 ± 7	4.6 × 10^3^ (13%)

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
