# Peer review of "Effects of Liposome and Cardiolipin on Folding and Function of Mitochondrial Erv1"

_ijms, 2020, doi:10.3390/ijms21249402_

Round 1

Reviewer 1 Report

The scientific article entitled “Effects of liposome and cardiolipin on folding and function of mitochondrial Erv1” reports results on the effect of cardiolipin-containing liposomes on the Erv1 oxidase and cytochrome c reductase activity. The authors present evidence that liposomes inhibit Erv1 cytochrome c reductase activity in a cardiolipin-concentration dependent manner and propose that, for Erv1 of yeast mitochondria, molecular oxygen is a better substrate than cytochrome c.

The manuscript is scientifically sound although several concerns need to be addressed:

The rate of non-enzyme catalysed cytochrome c reduction by DTT in the absence of Erv1 was also measured  in the presence of liposomes ?

Did the authors measure the effect of DTT (in the absence of Erv1) on oxygen consumption in the presence and absence of liposomes?

The molecular species (acyl chain composition) of cardiolipin are specific for tissues and organisms. Thus, the molecular species of cardiolipin used in the experiments should be indicated and discussed.

In addition to the ratio of phospholipids in liposomes, the ratio of phospholipids/Erv1 in the experiments should be better specified and discussed.

A thorough check of the text and references is recommended.

Fig 4 need to be better described.

Ref.s 16 and 24; 29 and 37; 21 and 38, are the same.

Ref.s 1 and 22, Journal name should be specified.

Lines 310-312  “The extinction coefficient 550 of 29.5 mM-1 cm-1 for oxidized cytochrome c and 8.4 mM-1 cm-1 for reduced cytochrome c were used”. Is this statement correct?

Line 320 “O2 concentration (~250 mM)” is not correct

Line 146: Change Fig 3 A in Fig 3

Line 217: Change acryl in acyl

Line 281: Change centrifuged in centrifugation

Author Response

The scientific article entitled “Effects of liposome and cardiolipin on folding and function of mitochondrial Erv1” reports results on the effect of cardiolipin-containing liposomes on the Erv1 oxidase and cytochrome c reductase activity. The authors present evidence that liposomes inhibit Erv1 cytochrome c reductase activity in a cardiolipin-concentration dependent manner and propose that, for Erv1 of yeast mitochondria, molecular oxygen is a better substrate than cytochrome c.

The manuscript is scientifically sound although several concerns need to be addressed:

The rate of non-enzyme catalysed cytochrome c reduction by DTT in the absence of Erv1 was also measured in the presence of liposomes?

  • Yes, and ‘In all cases, the rate of non-enzyme catalysed cytochrome c reduction by DTT in the absence of Erv1 was negligible’ was added to the text, at line 178-179

Did the authors measure the effect of DTT (in the absence of Erv1) on oxygen consumption in the presence and absence of liposomes?

  • Yes, and we added ‘As shown at the beginning of Figure 4A, there was no oxygen consumption by DTT in the absence of Erv1’ to the text (lines 149-150), and indicated when Erv1 was added in the Figure 4A.

The molecular species (acyl chain composition) of cardiolipin are specific for tissues and organisms. Thus, the molecular species of cardiolipin used in the experiments should be indicated and discussed.

  • We added following sentence at lines 103-104: The acyl chains of phospholipids of the yeast S cerevisiae are mainly C16:1 and C18:1 with minor amounts of C14:0, C16:0 and C18:0 [34]. Since C16:1 lipids are not commercially, accordingly, …

In addition to the ratio of phospholipids in liposomes, the ratio of phospholipids/Erv1 in the experiments should be better specified and discussed.

  • The molar ratios of lipids:Erv1 were 60:1 in figure 3 spectra measurement, and 100:1 in figure 4 kinetic study, which have been added to the corresponding figure legends and discussion (line 207).

Fig 4 need to be better described.

  • Apart from above, we also added Erv1 and liposome concentrations to Figure 4 legend.

A thorough check of the text and references is recommended.

Ref.s 16 and 24; 29 and 37; 21 and 38, are the same.

Ref.s 1 and 22, Journal name should be specified.

  • We have corrected all the reference mistakes, thanks!

Lines 310-312 “The extinction coefficient 550 of 29.5 mM-1 cm-1 for oxidized cytochrome c and 8.4 mM-1 cm-1 for reduced cytochrome c were used”. Is this statement correct?

  • The sentence has been revised to ‘The extinction co-efficient e at 550 nm of 29.5 mM-1cm-1 for oxidized cytochrome c and 8.4 mM-1cm-1 for reduced cytochrome c were used.’ (now lines 321- 322)

Line 320 “O2 concentration (~250 mM)” is not correct

  • Changed to μM (should be micromolar)

Line 146: Change Fig 3 A in Fig 3

  • A was deleted

Line 217: Change acryl in acyl

  • Changed to acyl

Line 281: Change centrifugation in centrifugation

  • Changed to centrifugation

Reviewer 2 Report

The manuscript "Effects of liposome and cardiolipin on folding and function of mitochondrial Erv1" (ijms-1017929) by Tang et al. suggests an inhibitory effect of cardiolipin-containing liposomes on Erv1 cytochrome c reductase activity. The Erv1 oxidase activity in the absence and presence of liposomes with various cardiolipin content was measured by oxygen consumption and cytochrome c reduction by spectroscopy. Whereas cardiolipin did not affect Erv1 oxidase activity, the protein's reduction activity on cytochrome c appeared to be inhibited in a concentration-dependent manner by cardiolipin. Unfortunately, the aim of the study is not well defined and the conclusions lack experimental support. Therefore, in its present state the manuscript is not recommended for publication in IJMS. 

Major comments

Fig. 5 B-D. The effects on cytochrome c reductase activity of Erv1 might be due to cytochrome c interacting with cardiolipin-containing liposomes (and not Erv1), which has been observed previously. It has also been suggested that cytochrome c interacts with the mitochondrial inner membrane.

Fig. 5 B-D. Have the authors tried to add cardiolipin by itself and not  cardiolipin-containing liposomes to see the effects of Erv1 on cytochrome c reductase activity?

Lines 209-210: "the membrane association of Erv1 in mitochondria are most likely due to its interaction with Mia40, rather than lipids". Therefore, the scope of this study strikes odd and does not make much sense.

Minor comments

Lines 146-147: "suggesting the presence of these liposomes did not affect the cofactor binding and thus folding of Erv1". Since the protein is supposedly already folded and FAD bound, maybe it would be better something like: "suggesting the presence of these liposomes did not affect Erv1 FAD binding or structure".

Line 217: "acryl tails". Maybe it should be "acyl chains".

Line 240: "(~3 M)". There is an unexplained cubic symbol before the "M".

Author Response

Major comments

Fig. 5 B-D. The effects on cytochrome c reductase activity of Erv1 might be due to cytochrome c interacting with cardiolipin-containing liposomes (and not Erv1), which has been observed previously. It has also been suggested that cytochrome c interacts with the mitochondrial inner membrane.

  • Yes, and here our study showed that such interactions inhibits the cytochrome c reductase activity of Erv1.

Fig. 5 B-D. Have the authors tried to add cardiolipin by itself and not  cardiolipin-containing liposomes to see the effects of Erv1 on cytochrome c reductase activity?

  • No, we did not test adding cardiolipin by itself, as it is not biologically relevant.

Lines 209-210: "the membrane association of Erv1 in mitochondria are most likely due to its interaction with Mia40, rather than lipids". Therefore, the scope of this study strikes odd and does not make much sense.

  • Please note that before this study, it was unknown whether or not the membrane association of Erv1 affects its function, e.g. oxidase and/or cytochrome c reductase activities. This is the first study addressing this question. One of our conclusion is: the membrane association of Erv1 in mitochondria are most likely due to its interaction with Mia40, rather than lipids

Reviewer 3 Report

The manuscript addresses the dual activity of Erv1, an essential enzyme present in the mitochondrial intermembrane space. Authors performed all experiments in vitro, using liposomes, mimicking the lipid composition of mitochondrial membranes. They showed that liposomes have a strong inhibitory effect on the cytochrome c reductase activity of Erv1, and that this inhibition is linked with the amount of cardiolipin inserted in liposomes.

The results are interesting, but they are not supported with a physiological effect.

I appreciate the work of the authors, it seems well done and the methods well explained. However, the work is too simple and poor in content. It's a good starting point to better investigate the physiological effect of Erv1, but further insights are needed.

In my opinion further experiments are necessary to consider the discovery in an applicative way.

Author Response

Yes, it is important to investigate the physiological effect of Erv1, but it is a big question beyond the scope of this study. This is the first study to address how the membrane association of Erv1 affects its function. The results should be interesting to researchers in the field to explore the physiological effect of Erv1 in future.

Round 2

Reviewer 3 Report

I understand the authors' response, however I expected to see an attempt for a physiological explanation in the discussion.
In any case, I remain of the idea that the work is too simple and not of a high enough scientific level to be published in this journal.

Author Response

We have tried our best to discuss the physiological implication in yeast in the last paragraph of our discussion (lines 244-253), and now we added following sentences in discussion (lines 253- 261):

‘The importance of the Erv1/ALR enzymes was also demonstrated by the identification of a disease directly related to a single mutation in ALR, the human homologue of Erv1 [41]. A single arginine-to-histidine substitution (R194H) in the C-terminus of ALR was identified in children with autosomal recessive myopathy; the effects included respiratory-chain deficiency, abnormal mitochondrial morphology and increased accumulation of mitochondrial DNA (mtDNA) deletions, a temperature-sensitive growth phenotype and reduced cytochrome c oxidase (complex IV) activity [41]. Protein characterization studies showed that the mutation results in decreased FAD-binding and thermal stability of the protein, and a mild effect on its function in vitro in the absence of liposomes [20, 42].’

It is followed by revised sentences ‘A study showed that the human short form of ALR (the Erv1 homologue) exhibits a higher cytochrome c reductase activity than its oxidase activity in the absence of liposomes, compared with yeast Erv1 [21]. Moreover, no yeast homologue of Ccp1 has been identified in human. It will therefore be interesting to interrogate how liposomes and cardiolipin affect the function of human ALR and more importantly the physiological effects. ...’